# Identification and Functional Analysis of Two Chitin Synthase Genes in the Common Cutworm, *Spodoptera litura*

**DOI:** 10.3390/insects11040253

**Published:** 2020-04-17

**Authors:** Hai-Zhong Yu, Ning-Yan Li, Yan-Xin Xie, Qin Zhang, Ying Wang, Zhan-Jun Lu

**Affiliations:** 1School of Life Sciences, Gannan Normal University, Ganzhou 341000, China; yuhaizhong1988@163.com (H.-Z.Y.); 18435203582@163.com (N.-Y.L.); 18970750806@163.com (Y.-X.X.); zhangqin0500092@163.com (Q.Z.); tianbian8802@163.com (Y.W.); 2National Navel Orange Engineering and Technology Research Center, Gannan Normal University, Ganzhou 341000, China

**Keywords:** chitin synthase, RNA interference, expression patterns, 20-hydroxyecdysone, *Spodoptera litura*

## Abstract

Chitin is one the main components of the insect cuticle, and chitin synthase (CHS) is an important enzyme required for chitin formation. CHS has been characterized in various insect species, but the structure and biochemical properties in *Spodoptera litura* have not been determined. In this study, we identified two *CHS* genes, *SlCHS1* and *SlCHS2*, which encode proteins with 1565 and 1520 amino acid residues, respectively. Transcriptional analysis suggested that *SlCHS1* has a high expression level in the integument whereas *SlCHS2* showed the highest expression level in the midgut. During *S. litura* growth and development, *SlCHS1* and *SlCHS2* were both predominantly expressed in the fourth-instar larval stage. In addition, the expression of *SlCHS1* and *SlCHS2* could be induced by 20-hydroxyecdysone (20E). Silencing of *SlCHS1* by RNA interference significantly inhibited the pupation and molting of *S. litura* larvae (RNAi), while knockdown of *SlCHS2* had no significant effects on the *S. litura* phenotype. These results may provide a new molecular target for control of *S. litura*.

## 1. Introduction

Chitin is a linear amino polysaccharide polymer made up of β-1,4-*N*-acetyl-d-glucosamine (GlcNAc) and is the second most abundant biological polymer after cellulose in nature. Chitin has been detected in a wide variety of organisms, ranging from the simplest algae to fungi, nematodes and arthropods [1,2]. In insects, chitin is essential for the structural integrity of the cuticle, and the peritrophic matrix (PM) lining the midgut epithelium, which protects the intestinal epithelium from mechanical disruption and invasion by various pathogens [3,4]. The balance of chitin content is essential for the growth and development (molting) of insects [5]. However, humans and higher animals appear to be completely devoid of chitin. Therefore, insecticides using chitin metabolism as targets can effectively control pests, but have no effect on humans. Chitin biosynthesis starts with the disaccharide trehalose, culminating in the polymerization of the *N*-acetyl glucosamine subunits by chitin synthase to produce chitin microfibrils [6]. In addition, this process requires various enzymes including hexokinase (HK), glucose-6-phosphate isomerase (G6PI), glutamine-fructose-6-phosphate aminotransferase (GFAT), glucosamine-6-phosphate *N*-acetyltransferase (GNPNA), phosphoglucosamine mutase (PAGM) and UDP-*N*-acetylglucosamine pyrophosphorylase (UAP). The last step is the chitin biosynthetic pathway catalyzed by chitin synthase (CHS), which can catalyze the transfer of sugar moieties from activated sugar donors to specific acceptors [7].

Chitin synthase (CHS, EC 2.4.1.16) is an integral membrane glycosyltransferase that is essential for chitin chain polymerization and deposition in insect chitinous structures [8]. CHS is a transmembrane protein that plays an important role in chitin synthesis [9]. *CHS* cDNA sequences have been characterized in several insect species, including *Drosophila melanogaster* [10], *Manduca sexta* [11], *Tribolium castaneum* [1], *Locusta migratoria* [12], *Spodopetra frugiperda* [13], and *Diaphorina citri* [14]. In insects, CHS are divided into two groups, namely, CHS1 and CHS2, based on domain composition, sequence homology, tissue localization and physiological role [15]. *CHS1* is exclusively expressed in the epidermis underlying the cuticular exoskeleton and related ectodermal cells such as tracheal cells, while *CHS2* is highly expressed in the midgut, and its coding enzyme is responsible for the synthesis of PM-associated chitin [16].

In recent years, RNA interference (RNAi) has been widely used to research the functions of *CHS* in different species. Chen et al. revealed that the cuticle of *Spodoptera exigua* larvae was disordered and that the epithelial walls did not expand uniformly after silencing *CHSA* [17]. In *Leptinotarsa decemlineata*, knockdown of *CHSAa*, *CHSAb* and *CHSB* in second- and fourth-instar larvae lowered chitin contents in whole body and integument samples and thinned tracheal taenidia [18]. However, the functions of CHS have not been reported in *Spodoptera litura*. RNAi refers to highly conserved cellular mechanisms in which Argonaute (Ago) family proteins bind small RNAs to trigger the degradation of longer RNAs through the sequence complementarity [19]. Delivery of dsRNA in insects can induce cascades of the RNAi pathway by which the enzyme Dicer2 cleaves dsRNA into fragments of siRNAs that bind to the RNA-induced silencing complex (RISC), which recognizes the target mRNA, binds to it and triggers degradation of the homologue endogenous mRNA [20,21]. Since the first discovery of this process in *Caenorhabditis elegans* by Fire et al., gene knockdown through RNAi induced by double stranded RNA (dsRNA) has been widely applied for the management of insect pests [22].

*Spodoptera litura* (Lepidoptera: Noctuidae) is an important herbivorous pest responsible for widespread economic damage to numerous field vegetables and ornamental plants in tropical and subtropical regions [23]. At present, control of *S. litura* is primarily achieved through the application of various chemical insecticides. However, *S. litura* has evolved high resistance to every class of pesticides used against it [24]. Shad et al. revealed that *S. litura* shows a high level of resistance to spinosad, indoxacarb, and methoxyfenozide [25]. In addition, many field populations of *S. litura* have developed resistance to multiple insecticides in South Asia, including chlorpyrifos, β-cypermethrin and methomyl [26]. Therefore, it is highly important to identify environmentally friendly methods to control *S. litura*. Chitin is found in insect exoskeletons, but is not found in vertebrates [27]. Therefore, we considered using chitin metabolism-related genes as targets to control *S. litura*.

In the current study, two cDNAs encoding chitin synthase (*SlCHS1* and *SlCHS2*) were identified from the genome database of *S. litura*, and their spatial and temporal expression profiles were analyzed. In addition, the expression levels of *SlCHS1* and *SlCHS2* can be induced by 20E. Furthermore, silencing of *SlCHS1* influences *S. litura* larvae pupation and molting. However, silencing of *SlCHS2* has no significant influence for molting of *S. litura*. Herein we explore whether these genes can provide useful information on CHSs as targets for the identification of novel insecticides?

## 2. Materials and Methods

### 2.1. Spodoptera Litura Rearing and Tissue Preparation

*S. litura* larvae were collected from the orange orchard at the National Navel Orange Engineering Research Center (NORC), Gannan Normal University, Ganzhou, China. Larvae were reared in culture dishes on an artificial diet at 27 °C and 70%–75% relative humidity, with a photoperiod of 12 h light and 12 h dark, until they became adult moths. The main components of the artificial diet include corn starch, soybean flour, agar, yeast powder, sorbic acid and cholesterol. All male and female adults were placed in a plastic case, and moderate hydromel was added to keep the plural adults alive. The produced eggs were reared based on the above conditions. *S. litura* were collected at different developmental stages, including larvae, pupae and adults. Moreover, the first day of sixth-instar larvae were dissected to obtain various tissues, including the integument, head, Malpighian tubule, fat body and midgut. The midgut was cleaned using precooled DEPC-water to remove the remaining food debris, and stored at −80 °C.

20E treatment was performed according to a previous report with some modifications [28]. In brief, a total of 2 μg of 20E was dissolved in 4 μL of dimethyl sulfoxide (DMSO) to prepare the working solution and then injected into larvae on the first day their sixth instar. DMSO was injected into other first day, sixth-instar larvae, as a control. The midgut and integument samples were collected after 1, 12, 36 and 48 h, and stored at −80 °C. Each treatment was repeated with three biological replicates.

### 2.2. RNA Isolation and cDNA Synthesis

To analyze the spatiotemporal expression patterns of *SlCHS1* and *SlCHS2*, *S. litura* total RNA was extracted from different tissues of sixth-instar larvae (integument, head, Malpighian tubule, fat body and midgut) and at different developmental stages (second-instar, third-instar, fourth-instar, fifth-instar, and sixth-instar larvae, and pupae and adults) using the animal tissue total RNA kit (Simgen, Hangzhou, Zhejiang, China). RNA concentration and purity were assayed using a NanoDrop2000 spectrophotometer (Thermo Fisher Scientific, New York, NY, USA) at absorbance ratios of A_260/230_ and A_260/280_. The integrity of total RNA was confirmed using standard agarose gel electrophoresis with ethidium bromide (EB) staining. Total RNA was reverse-transcribed in a 20 μL reaction system using a Fast 1st strand cDNA Synthesis kit (with gDNase) (Simgen) according to the manufacturer’s instructions. In brief, 2.0 μL of 5 × gRNA buffer and 1 μg of total RNA were mixed, and then RNase-free water was added to reach 10 μL, which was then incubated at 42 °C for 3 min. Afterward, 4 μL of 5 × RT buffer and 2 μL of RT enzyme mix were added and incubated at 95 °C for 3 min. The cDNA was stored at −20 °C for later use.

### 2.3. Sequencing Analysis of SlCHS1 and SlCHS2

To identify chitin synthase genes in *S. litura*, a TBLASTIN search of the *S. litura* genome database (https://www.ncbi.nlm.nih.gov/genome/14271?genome_assembly_id = 350050) was performed using the amino acid sequences of *D. melanogaster CHS1* (NM_079509.3), *D. melanogaster CHS2* (NM_079485.4), *M. sexta CHS1* (AY062175.2) and *M. sexta CHS2* (AY821560.1) as queries. This resulted in the identification of two cDNA sequence that we have named as *SlCHS1* and *SlCHS2*. To confirm the correctness of the two candidate *CHS* sequences, reverse transcription PCR (RT-PCR) was carried out to amplify the full-length ORF sequence using gene-specific primers (Table 1). The deduced amino acid sequences of *SlCHS1* and *SlCHS2* were analyzed by using DASTAR software. The open reading frame (ORF) was predicted according to the ORF finder tool (https://www.ncbi.nlm.nih.gov/orffinfer/). The molecular weight (MW) and isogenic point (pI) of SlCHS1 and SlCHS2 were calculated using ExPASy (http://web.expasy.org/compute_pi). The signal peptides of SlCHS1 and SlCHS2 were predicted using SignalP 4.1 Server (http://www.cbs.dtu.dk/services/SignalP). The functional domain was predicted by using SMART software (http://smart.embl-heidelberg.de/). The membrane-spanning domain was predicted by TMHMM Server v. 2.0 (http://www.cbs.dtu.dk/services/TMHMM/). The phylogenetic tree was constructed with MEGA 7.0 software using the neighbor-joining method with 1000-fold bootstrap resampling. Protein sequences from different insect species were obtained from GenBank (http://www.ncbi.nlm.nih.gov/) and used in the phylogenetic analysis (Appendix A).

### 2.4. RT-qPCR Analysis of SlCHS1 and SlCHS2 Expression Levels

RT-qPCR was performed to analyze the relative expression levels of *SlCHS1* and *SlCHS2*. The primers were designed using Primer Premier 5.0 software (Table 1). The 20-µL reaction mixture for RT-qPCR contained 10 µL of SYBR II, 8 µL of ddH_2_O, 0.5 µL of forward primer, 0.5 µL of reverse primer, and 1.0 µL of cDNA template. The thermal cycling profile consisted of an initial denaturation at 95 °C for 60 s and 40 cycles of 95 °C for 10 s, 60 °C for 10 s, and 72 °C for 10 s. The reactions were performed with a LightCycle^®^ 96 PCR Detection System (Roche, Basel, Switzerland). Relative expression levels were calculated by using the 2^−∆∆Ct^ method. There were three biological replicates and three technique replicates for each sample. The reference gene chosen for analysis of *SlCHS1* and *SlCHS2* in different tissues and different developmental stages was *glyceraldehyde-3-phosphate dehydrogenase* (*GAPDH*). All data were analyzed using a one-way analysis of variance (ANOVA) and Tukey’s test.

### 2.5. dsRNA Synthesis and RNAi Analysis

dsRNA targeting *SlCHS1* and *SlCHS2* was synthesized using the T7 RioMAX Express RNAi System (Promega, San Luis Obispo, CA, USA) based on the manufacturer’s instructions. The forward and reverse primers were designed to amplify 439 bp and 421 bp for *SlCHS1* and *SlCHS2*, respectively (Table 1). GFP dsRNA was used as a control. The final concentration of dsRNA for injection was adjusted to 300 ng/µL and 500 ng/µL using DEPC water as the working solution. To ensure the RNAi efficiency, pre-pupal *S. litura* larvae were injected with 10 µL of *SlCHS1* and *SlCHS2* dsRNA using a microinjector (Sangon Biotech, Shanghai, China), and the midgut and integument were collected at 24 h and stored at -80 °C. After 24 h, all live insects were collected to isolate total RNA and synthesize cDNA. The effect of dsSlCHS1 and dsSlCHS2 on gene expression was evaluated by RT-qPCR. A total of three biological replicates were used for each experiment. All data were analyzed using ANOVA and Tukey’s test.

## 3. Results

### 3.1. cDNA and Deduced Amino Acid Sequences of SlCHS1 and SlCHS2

In total, two chitin synthase genes, *SlCHS1* and *SlCHS2* were identified from *S. litura*. The cDNA sequence of *SlCHS1* (GenBank accession number: XM_022964624.1) contains an ORF of 4698 bp encoding a protein of 1565 amino acid residues with a predicted MW of 178.1 kDa and a pI of 6.56. *SlCHS2* (GenBank accession number: XM_022821184.1) contains an ORF of 4563 bp encoding a protein of 1520 amino acid residues with a predicted MW of 174.1 kDa and a pI of 5.83. In terms of protein structure, SlCHS1 contains three domains, including an N-terminal domain (residues 1-561) with nine transmembrane helices, a putative catalytic domain (residues 562-900) and a C-terminal domain (residues 901-1565) with an additional seven transmembrane helices (Appendix A). SlCHS2 also has three domains, including an N-terminal domain (residues 1-552) with nine transmembrane helices, a putative catalytic domain (residues 553-891) and a C-terminal domain (residues 892-1520) with an additional seven transmembrane helices (Appendix A). In addition, both SlCHS1 and SlCHS2 contain signature sequences (EDR and QRRRW). By using NetNGlyc 1.0 software analysis, SlCHS1 contains six glycosylation sites, and SlCHS2 contains five glycosylation sites, suggesting that these proteins are glycosylated. The deduced amino acid sequences of these two chitin synthases exhibited seven highly conserved motifs (Figure 1). Based on the amino acid sequences of CHS from different insect species, a phylogenetic tree was generated using MEGA 5.0 to investigate the evolutionary relationship of SlCHS1 and SlCHS2 among the selected insect species. Insect CHSs can be divided into two classes, CHS1 and CHS2. The results showed that both SlCHS1 and SlCHS2 maintained high identity to *S. exigua* CHS1 and CHS2, respectively (Figure 2).

### 3.2. Spatiotemporal Expression Patterns of SlCHS1 and SlCHS2

The expression profiles of *SlCHS1* and *SlCHS2* in different developmental stages and different tissues were investigated by RT-qPCR. For tissue expression analysis, the first day of sixth-instar larvae were dissected to obtain different tissues, including the integument, head, Malpighian tubules, fat body and midgut, and the results suggested that both *SlCHS1* and *SlCHS2* were expressed in all *S. litura* tissues. Notably, *SlCHS1* was predominantly expressed in the integument followed by the head. Low expression of *SlCHS1* was detected in the Malpighian tubules. The expression level of *SlCHS1* in the integument was 380.2 times that in the Malpighian tubules, and its expression in the head was 191.7 times that of the Malpighian tubules. However, *SlCHS2* had the highest expression level in the midgut, whereas it showed low expression in the integument, head and fat body. The expression level of *SlCHS2* in the midgut was 4687.5 times that in the integument (Figure 3). For developmental stage expression analysis, the expression levels of *SlCHS1* and *SlCHS2* increased from the second-instar larvae to the fourth-instar larval stage and then decreased from the fourth-instar larvae to the sixth-instar larval stage. The *SlCHS1* expression level showed a significant fluctuation at the pupal stage, while the *SlCHS2* expression level had no significant change from the sixth-instar larval stage to the adult stage (Figure 3).

### 3.3. Analysis of the Expression Levels of SlCHS1 and SlCHS2 after 20E Treatment

20E is the key factor that controls the metamorphosis of insects. To analyze whether 20E regulates the transcriptional expression of *SlCHS1* and *SlCHS2*, 20E was injected into sixth-instar larvae. The results revealed that 20E can induce the expression levels of *SlCHS1* and *SlCHS2* (Figure 4). The expression level of *SlCHS1* showed no significant change between 20E treatment group and the control at 1 h. However, the *SlCHS1* expression level was significantly upregulated at 12 h after 20E treatment compared with the control group and then sharply decreased from 12 h to 36 h. The expression level of *SlCHS2* had no significant change from 1 h to 12 h after 20E treatment. However, compared with the control group, the *SlCHS2* expression level was significantly upregulated from 36 h to 48 h. These results indicated that the expression levels of *SlCHS1* and *SlCHS2* were affected by 20E.

### 3.4. RNAi-Based Silencing of SlCHS1 and SlCHS2 and Phenotype Analysis

To investigate the effect of *SlCHS1* and *SlCHS2* on *S. litura* molting, RNAi was performed by injection of dsRNA. The silencing efficiency was scrutinized in different concentrations of dsRNA by RT-qPCR. The results showed that neither *SlCHS1* nor *SlCHS2* could be silenced effectively at 24 h after injection of 300 ng/μL dsRNA. However, the expression levels of *SlCHS1* and *SlCHS2* were significantly (*p* < 0.01) reduced after injection of 500 ng/μL dsRNA at 24 h compared with the dsGFP control treatment (Figure 5A,B). After injection of dsSlCHS1, the *S. litura* larvae could not develop into pupae normally, and the adults were unable to molt completely. However, the larvae treated with dsSlCHS2 were not affected and pupated normally (Figure 5C). These results suggest that *SlCHS1* might play an important role during *S. litura* larval molting, while *SlCHS2* has no significant effect on *S. litura* molting.

## 4. Discussion

*Spodoptera litura* is one of the major agricultural pests worldwide and can seriously damage some important economic crops, including soybean, cotton, tobacco and cruciferous vegetables [29]. To date, the control of *S. litura* has been mainly dependent on chemical insecticides [30]. However, the long-term use of these insecticides not only pollutes the environment and affects human health, but also leads to the development of insect resistance against certain pesticides, including organophosphate and some biogenic insecticides [31,32]. Therefore, it is critical to find environmentally friendly insecticides for the control of *S. litura*. In recent years, some key genes from *S. litura* were identified as potential targets for the control of *S. litura*. Ji et al. revealed that silencing of *S. litura nicotinamide adenine dinucleotide phosphate (NADPH)-cytochrome P450 reductases* (*SlCPRs*) increased larval mortality by 34.6% (LC_15_ dose) and 53.5% (LC_50_ dose) by RNAi [33]. Wang et al. also found that RNAi-mediated silencing of *S. litura cytochrome P450 monooxygenases 321B1* (*SlCYP321B1*) further increased mortality by 25.6% and 38.9% when fifth-instar larvae were exposed to chlorpyrifos and *β*-cypermethrin [26]. However, these target genes also exhibited some defects because of insect insecticide resistance. In contrast, chitin forms the insect exoskeleton, plays important roles in physiological systems, and provides physical, chemical and biological protection. In the present study, we identified two chitin synthase genes from the *S. litura* genome database. The structural domain analysis suggested that both SlCHS1 and SlCHS2 contain 16 transmembrane helices (Appendix A). Depending on the number of predicted transmembrane helices, the N-terminus faces either the extracellular space or the cytoplasm. However, the C-terminal region is predicted to face the extracellular space and may be involved in protein– protein interactions or oligomerization [34]. Additionally, we also found that both SlCHS1 and SlCHS2 contain two typical chitin synthase signature motifs, including EDR and QRRRW, which are essential for the catalytic mechanism [35]. Moreover, SlCHS1 and SlCHS2 contain six and five glycosylation sites, respectively, suggesting that these proteins are glycosylated. Protein glycosylation is the covalent attachment of an oligosaccharide chain to a protein backbone and is considered to be a very common protein modification [36]. These results indicated that *S. litura* CHS plays an important role in chitin synthesis.

To further investigate the functions of *SlCHS1* and *SlCHS2*, we analyzed their expression patterns in different tissues and at different developmental stages, as well as 20E treatment. The results showed that *SlCHS1* had high expression in the integument and head, while *SlCHS2* had high expression in the midgut (Figure 3). In insects, chitin functions as a scaffold material, supporting the cuticle of the epidermis [37]. In *S. exigua*, northern blot analysis also revealed that *SeCHSA* is transcribed preferentially in the cuticle and tracheae [38]. Qu et al. revealed that the *Ostrinia furnacalis chitin synthase A* (*OfCHSA*) transcript was preferentially expressed in the epidermis [39]. In addition, the localization of *SlCHS1* was confirmed in the integument and midgut. Therefore, we speculated that *SlCHS1* may play a critical role in the process of cuticle formation. The insect midgut epithelium is commonly lined by an invertebrate-unique structure, the peritrophic matrix (PM), which facilitates the digestion of food and the protection of the gut epithelium [40]. We considered that *SlCHS2* may be involved in chitin formation in PM. In *T. castaneum*, the *CHSB* gene is the major or sole contributor to PM chitin synthesis [41]. At different developmental stages, both *SlCHS1* and *SlCHS2* showed high expression in the fourth-instar larval stage (Figure 3). In *Ectropis oblique*, *CHSA* expression was the strongest in third- and fourth-instar larvae [42]. However, the *SlCHS1* expression level showed a significant fluctuation at the pupal stage. During the growth and development of *O. furnacalis*, *OfCHSA* was mainly expressed during larval–larval molting and larval–pupal transformation [39]. Therefore, we speculated that *OfCHSA* also plays an important role in larval–pupal transformation. 20E plays critical roles in insect development and binds to the nuclear receptor heterodimer ecdysone receptor (EcR)/ultraspiracle (USP) [43]. In insects, the process of chitin biosynthesis is strictly coordinated with the cycle of molts. Thus, chitin biosynthesis may be regulated by 20E [15,44]. In this study, we found that both *SlCHS1* and *SlCHS2* expression levels can be induced after 20E treatment (Figure 4). In a previous report, analysis of mRNA levels showed that *BmCHSA-2b* was responsive to 20E [45]. In *Bactrocera dorsalis*, 20E induced the expression of *BdCHS1* and its variants [46]. Based on these results, we speculated that 20E can bind EcR and USP to form the ligand–receptor complex 20E-EcR/USP, and the receptor complex directly activates *SlCHS1* and *SlCHS2* expression thereafter.

RNAi has already proven its usefulness in functional genomics research in insects, but it also has considerable potential for the control of pest insects [47]. For the effective silencing of target genes, it is critical to transmit dsRNA into the body of insects to disrupt the expression of target genes [48]. Mechanistic studies have shown that double-stranded ribonucleases (dsRNases), endosomal entrapment, deficient function of the core machinery, and inadequate immune stimulation, contribute to limited RNAi efficiency [49]. Compared with the Coleoptera, Lepidoptera insects appear to have a low RNAi efficiency because dsRNAs were degraded more easily by RNase [50]. To ensure the RNAi efficiency, prepupa *S. litura* larvae were used for RNAi analysis. The results showed that the transformation of larval–pupae and pupae–adults was disrupted after silencing of *SlCHS1*, while *S. litura* larvae showed no obvious change after silencing of *SlCHS2* (Figure 5). In *T. castaneum*, *TcCHS1*-specific RNAi disrupted all three types of molt (larval–larval, larval–pupal and pupal–adult), while *TcCHS2*-specific RNAi had no effect on metamorphosis [41]. In *S. exigua*, the cuticle was disordered, and the epithelial walls of larval tracheae did not expand uniformly after injection of dsSeCHSA [21]. These results demonstrated that *SlCHS1* was mainly involved in the formation of epidermal structure and that *SlCHS2* was associated with PM formation.

## 5. Conclusions

Two cDNA sequences of *SlCHS1* and *SlCHS2* were identified based on genome database of *S. litura*. RT-qPCR analysis suggested that *SlCHS1* and *SlCHS2* were highly expressed in the integument and midgut, respectively. Developmental stage expression analysis showed that both *SlCHS1* and *SlCHS2* were predominantly expressed in the fourth-instar larval stage. In addition, the expression of *SlCHS1* and *SlCHS2* could be induced by 20E. Furthermore, silencing of *SlCHS1* significantly inhibited the pupation and molting of *S. litura* larvae by RNAi. These results suggest that *SlCHS1* and *SlCHS2* were involved in the formation of chitin in integument and peritrophic matrix (PM), respectively. In our following study, we will design some inhibitors directly targeted at *S. litura* chitin synthases and combined with RNAi to control *S. litura*.

## Figures and Tables

**Figure 1 insects-11-00253-f001:**
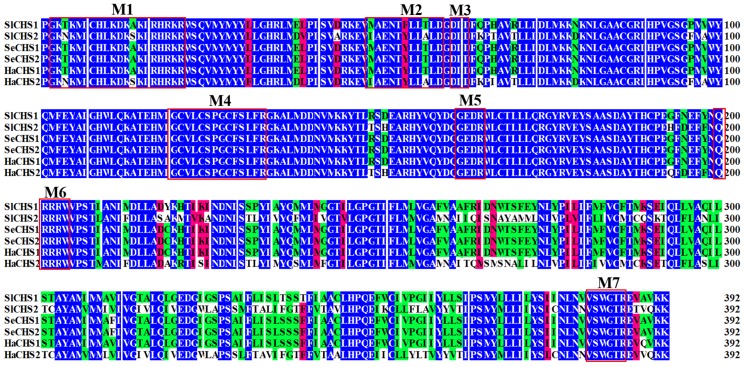
Multiple sequence alignment of the conserved catalytic domain of the chitin synthases (CHSs) from three insect species. CHSs are from *Spodoptera litura* (Sl), *Spodoptera exigua* (Se) and *Helicoverpa armigera* (Ha). Seven characteristic motifs (M 1–7) for insect chitin synthases are indicated with red boxes.

**Figure 2 insects-11-00253-f002:**
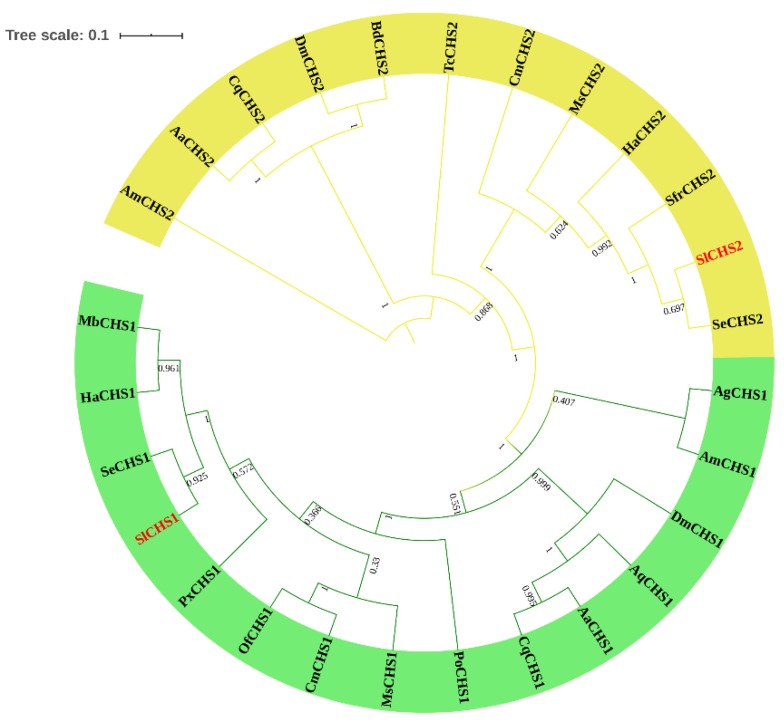
Phylogenetic relationship analysis of chitin synthases (CHSs) in different insect species. A phylogenetic tree was constructed using MEGA 5.0 software using the neighbor-joining method with a bootstrap value of 1000. CHSs are from *Spodoptera litura* (Sl), *Spodoptera exigua* (Se), *Spodoptera frugiperda* (Sfr), *Helicoverpa armigera* (Ha), *Manduca sexta* (Ms), *Cnaphalocrocis medinalis* (Cm), *Tribolium castaneum* (Tc), *Apis mellifera* (Am), *Aedes aegypti* (Aa), *Culex quinquefasciatus* (Cq), *Drosophila melanogaster* (Dm), *Bactrocera dorsalis* (Bd), *Aphis glycines* (Ag), *Anopheles quadrimaculatus* (Aq), *Ostrinia furnacalis* (Of), *Plutella xylostella* (Px), *Mamestra brassicae* (Mb) and *Phthorimaea operculella* (Po).

**Figure 3 insects-11-00253-f003:**
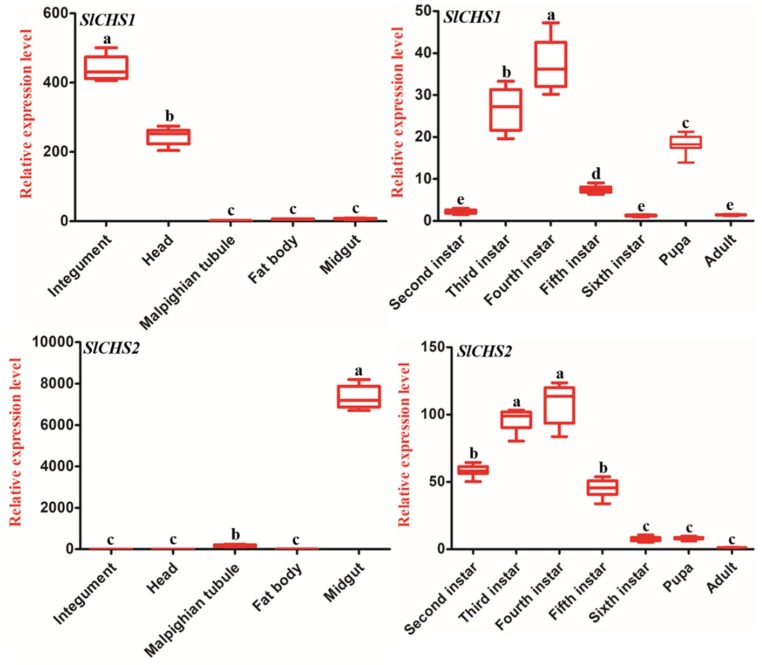
Expression patterns of *SlCHS1* and *SlCHS2* in different tissues of sixth-instar larvae and at different developmental stages of the larvae of *S. litura*. Relative mRNA levels of *SlCHS1* and *SlCHS2* as examined using RT-qPCR. Data were normalized using *glyceraldehyde-3-phosphate dehydrogenase* (*GAPDH*) and are represented as the means ± standard errors of the means from three independent experiments. Relative expression levels were calculated using the 2^−∆∆Ct^ method. Statistical analysis was performed using SPSS software. All data were analyzed using ANOVA and Tukey’s test. The significant differences are indicated by a different letter, for example, a, b, and c (*p* < 0.05).

**Figure 4 insects-11-00253-f004:**
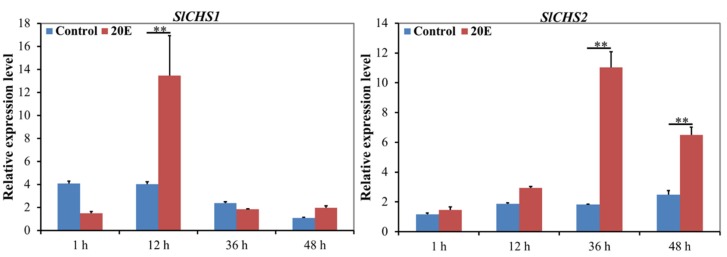
Expression levels of *SlCHS1* and *SlCHS2* after 20E treatment in *S. litura*. Data were normalized using *glyceraldehyde-3-phosphate dehydrogenase* (*GAPDH*) and are represented as the means ± standard errors of the means from three independent experiments. Relative expression levels were calculated using the 2^−∆∆Ct^ method. Statistical analysis was performed using SPSS software. All data were analyzed using ANOVA and Tukey’s test. Significant differences are indicated by ** (*p* < 0.01).

**Figure 5 insects-11-00253-f005:**
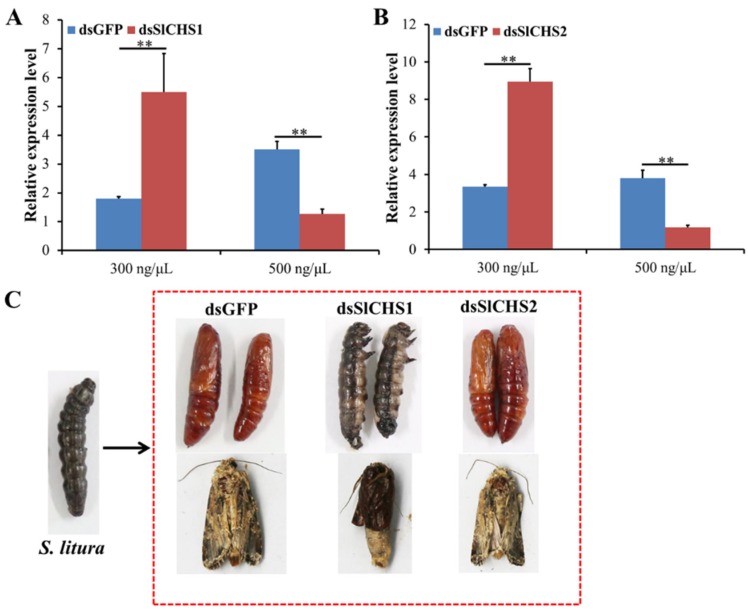
Effect on *S. litura* after RNA interference of *SlCHS1* and *SlCHS2*. (**A**) Relative expression levels of *SlCHS1* when *S. litura* was treated with dsSlCHS1 and dsGFP at different concentrations. **(B**) Relative expression levels of *SlCHS2* when *S. litura* was treated with dsSlCHS2 and double-stranded dsGFP at different concentrations. The mean expression level is represented for three biological replicates. Relative expression levels were calculated using the 2^−∆∆Ct^ method. Statistical analysis was performed using SPSS software. All data were analyzed using ANOVA and Tukey’s test. Significant differences are indicated by ** (*p* < 0.01). (**C**) Representative phenotypes of *S. litura* at 24 h after dsGFP, dsSlCHS1 and dsSlCHS2 treatment.

**Table 1 insects-11-00253-t001:** Primers used in this study.

Primers	Sequences	Purpose
SlCHS1-F	ATGGCGACGTCAGGAGGGA	ORF amplify
SlCHS1-R	TTAGAATCTACCCTGGAAGGAAAC	
SlCHS2-F	ATGGCGAGACAAAGAACTTTAAG	
SlCHS2-R	TCACGCGAAATGGTCCGAG	
SlCHS1-RT-F	TCACCGACTAATGGAACTGCC	RT-qPCR
SlCHS1-RT-R	ACCACACCATAGGACCAGAGCC	
SlCHS2-RT-F	CCCTGGATGCTTCTCCCTCTT	
SlCHS2-RT-R	CGTTGGTTGAAGAACTCGTCG	
GAPDH-F	GGGTATTCTTGACTACAC	
GAPDH-R	CTGGATGTACTTGATGAG	
ds-SlCHS1-F	GGATCCTAATACGACTCACTATAGGGAAGAACAAGAATCTGGGAGC	dsRNA synthesis
ds-SlCHS1-R	GGATCCTAATACGACTCACTATAGGGATAGTGTGTTTGTAATCGGCA	
ds-SlCHS2-F	GGATCCTAATACGACTCACTATAGGAAAAGGCGACTGAACACAT	
ds-SlCHS2-R	GGATCCTAATACGACTCACTATAGGAAGATTGTACCAGGACCCA	
ds-GFP-F	GGATCCTAATACGACTCACTATAGGCAGTGCTTCAGCCGCTACCC	
ds-GFP-R	GGATCCTAATACGACTCACTATAGGACTCCAGCAGGACCATGTGAT	

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
