# Peer review of "Identification and Functional Analysis of Two Chitin Synthase Genes in the Common Cutworm, Spodoptera litura"

_insects, 2020, doi:10.3390/insects11040253_

Round 1

Reviewer 1 Report

Here, authors present their study entitled "Identification and Functional Analysis of Two Chitin Synthase Genes in the Common Cutworm, Spodoptera litura". Overall, the manuscript is well written and findings are well described. I attach some minor comments:

line 13: Add a brief statement of why CHS are important to study in insects

line 15: ... biochemical properties "in" Spodoptera litura...?

Lines 56-70: I would rather start this paragraph with examples of CHS in other moths, and what happens when silenced by RNAi. Then, describe briefly what is RNAi and how useful.

line 87: Maybe it would provide useful information of CHSs as targets for the identification of novel insecticides.

line 123: Why D. citri?

line 131: Did authors tested phylogeny using maximum likelihood? It would worth the try.

line 136: Which conditions?

line 195: I believe that Figure 1 could be more appropiate for Supp Info. Instead, I suggest to include here a multiple sequence alignment showing those motifs conserved in CHSs.

line 209: Although the tree seems fine to me. I would suggest to run ML method to test the same relationships. Likewise, I suggest to edit the tree (commonly in Newick format) using FigTree software or iTOOL server, for a better looking tree.

line 232: Why relative expression is showed differently between tissue and developmental stage?. I suggest to use bars for both.

line 241: Please, explain briefly why authors selected sixth instar.

lines 360-361: I suggest to extend a little bit more this section, explaining how the study of these genes could impact on S. litura control, and what other experiments should follow.

Author Response

  1. Line 13: Add a brief statement of why CHS are important to study in insects

Reply: Thanks for reviewer’s valuable comments. We have revised the related descriptions in previous manuscript, seeing Line 14 to Line 16.

  1. Line 15:….biochemical properties “in” Spodoptera litura…?

Reply: Thanks for reviewer’s thoughtful comments. We have revised “of” into “in” in previous manuscript, seeing Line 17.

  1. Lines 56-70: I would rather start this paragraph with examples of CHS in other moths, and what happens when silenced by RNAi. Then, describe briefly what is RNAi and how useful.

Reply: Thanks for reviewer’s valuable comments. We have revised the related descriptions in previous manuscript, seeing Line 58 to Line 81.

  1. line 87: Maybe it would provide useful information of CHSs as targets for the identification of novel insecticides.

Reply: Thanks for reviewer’s thoughtful suggestions. We have revised the related descriptions in previous manuscript, seeing Line 99 to Line 101.

  1. line 123: Why D. citri?

Reply: We are sorry for the incorrect descriptions. We have revised “D. citri” into “S. litura” in previous manuscript, seeing Line 137.

  1. line 131: Did authors tested phylogeny using maximum likelihood? It would worth the try.

Reply: Thanks for reviewer’s thoughtful suggestions. We have tested phylogeny using maximum likelihood, the result was added to attachment.

  1. line 136: Which conditions?

Reply: We are sorry for unclear descriptions in previous manuscript. The conditions indicated that the relative expression level was analyzed in different tissues, different developmental stages and 20E treatment. We have deleted the unclear descriptions, seeing Line 157 to Line 158.

  1. line 195: I believe that Figure 1 could be more appropriate for Supp Info. Instead, I suggest to include here a multiple sequence alignment showing those motifs conserved in CHSs.

Reply: Thanks for reviewer’s thoughtful and valuable suggestions. We have added the Figure 1 and Figure 2 to Supplementary information and supplement the multiple sequence alignment analysis, seeing Line 217.

  1. line 209: Although the tree seems fine to me. I would suggest to run ML method to test the same relationships. Likewise, I suggest to edit the tree (commonly in Newick format) using FigTree software or iTOOL server, for a better looking tree.

Reply: Thanks for reviewer’s valuable suggestions. We have tested phylogeny using maximum likelihood, the result was added to attachment.

  1. line 232: Why relative expression is showed differently between tissue and developmental stage? I suggest to use bars for both.

Reply: Thanks for reviewer’s valuable comments. In this study, the RT-qPCR results showed that SlCHS1 had a high expression in the integument, while SlCHS2 had a high expression in the midgut. Due to different functions of SlCHS1 and SlCHS2 in the developmental of insect, we considered that SlCHS1 might be involved in the formation of chitin in the integument, while SlCHS2 might be associated with the formation of chitin in the peritrophic membrane (PM). The peritrophic membrane is composed of chitin and proteins, of which peritrophins are the most important. It is proposed here that, during evolution, midgut cells initially synthesized chitin and peritrophins derived from mucins by acquiring chitin-binding domains, thus permitting the formation of PM (The origine and function of insect peritrophic membrane and peritrophic gel). In Tribolium castaneum, the chitin synthase genes TcCHS1 and TcCHS2 were involved in synthesis of chitin in epidermal cuticle and midgut peritrophic matrix (The Tribolium chitin synthase genes TcCHS1 and TcCHS2 are specialized for synthesis of epidermal cuticle and midgut peritrophic matrix). According to these results, we speculated that SlCHS1 and SlCHS2 have different functions. In addition, we have revised the Figure 3, seeing Line 261.     

  1. line 241: Please, explain briefly why authors selected sixth instar.

Reply: Thanks for reviewer’s thoughtful comments. The Spodoptera litura sixth-instar is the critical transition period from larva to pupa. During this time, the larvae enter a critical stage of growth and development. The change in body shape is very noticeable. The larval integument and midgut enter a period of rapid growth. Therefore, we selected sixth-instar for 20E treatment.     

  1. lines 360-361: I suggest to extend a little bit more this section, explaining how the study of these genes could impact on S. litura control, and what other experiments should follow.

Reply: Thanks for reviewer’s thoughtful suggestions. We have added the related descriptions in previous manuscript, seeing Line 393 to Line 397.

Reviewer 2 Report

L85-86: What does the following imply? ‘However, this silencing had no significant effect after knockdown of SlCHS2.’? Please elaborate on the implications of this finding.

L94: ‘…case, and moderate hydromel was added to keep the adult alive.’ Change adult to plural adults.

L94-96: This sentence has grammatical issues, “The produced eggs were reared 95 using an artificial diet based on the above conditions, and then S. litura larvae at different 96 developmental stages, pupae and adults were collected.”

L115: what cDNA synthesis kit?

L220: typo ‘Norably’ should be Notably.

L353: what is PM?

L358: Please elaborate on why/if fourth-instar larval stage is a key stage/relevant stage for pest-control.

General comments:

If SlCHS2 is primarily found in the midgut (which is initially elucidated from the spatiotemporal expression patterns analyses), it seems natural that it would not influence molting. The manuscript can be better written to follow this logical line.

The final paragraph of the introduction needs to be re-written for improved clarity (L81-87), in particular, the following sentence (L85-86): “However, this silencing had no significant effect after knockdown of SlCHS2.”

This is confusing because in the results it says the following (L262-266): “After injection of dsSlCHS1, the S. litura larvae could not transform into pupa normally. In addition, the adults could not moult completely. However, the larvae could transform into pupae and adults after silencing SlCHS2 compared with the control groups (dsGFP treatment) (Figure 6C). These results suggested that knockdown of SlCHS1 inhibits the synthesis of chitin and influence S. litura moulting, while SlCHS2 has no significant effect on S. litura moulting.”

As written L85-86 would imply that knocking down SlCHS2 somehow negates the effects of silencing SlCHS1, which does not seem to be what the results are stating. I suspect this is a wording issue.

Finally, the developmental stage analyses don’t seem to tie back to the objectives of the work. Are certain stages more important for pest control? How does this relate their fourth instar findings? SlCHS1 and SlCHS2 follow the same pattern, so I am not sure what the take away is.

Author Response

  1. L85-86: What does the following imply? ‘However, this silencing had no significant effect after knockdown of SlCHS2.’? Please elaborate on the implications of this finding.

Reply: We are sorry for unclear descriptions in previous manuscript. We have revised the related descriptions, seeing Line 98 to Line 99.

  1. L94: ‘…case, and moderate hydromel was added to keep the adult alive.’ Change adult to plural adults.

Reply: Thanks for reviewer’s thoughtful comments. We have revised “adult” to “plural adults” in previous manuscript, seeing Line 109.

  1. L94-96: This sentence has grammatical issues, “The produced eggs were reared 95 using an artificial diet based on the above conditions, and then S. litura larvae at different 96 developmental stages, pupae and adults were collected.”

Reply: Thanks for reviewer’s valuable comments. We have revised the grammatical in previous manuscript, seeing Line 109 to Line 112.

  1. L115: what cDNA synthesis kit?

Reply: Thanks for reviewer’s thoughtful comments. We have added the detailed information for cDNA synthesis kit, seeing Line 130.

  1. L220: typo ‘Norably’ should be Notably.

Reply: Thanks for reviewer’s thoughtful suggestions. We have revised “Norable” to “Notably”, seeing Line 249.

  1. L353: what is PM?

Reply: Thanks for reviewer’s thoughtful comments. The PM indicates peritrophic matrix.

  1. L358: Please elaborate on why/if fourth-instar larval stage is a key stage/relevant stage for pest-control.

Reply: Thanks for reviewer’s thoughtful and valuable comments. In this study, RT-qPCR results showed that SlCHS1 and SlCHS2 had a high expression in the fourth-instar larval stage. We considered that the fourth-instar is a critical stage for growth and development of S. litura. Before the fourth-instar stage, the growth of S. litura larvae is relatively slow to satisfy the development of various tissues, and then the larvae enter the glutonous stage. During this stage, the integument and peritrophic matrix enter a rapid growth stage. Therefore, the S. litura can promote chitin synthesis by increasing the expression of SlCHS1 and SlCHS2 to maintain the structure of integument and peritrophic matrix. However, the specific reasons need further research.       

  1. If SlCHS2 is primarily found in the midgut (which is initially elucidated from the spatiotemporal expression patterns analyses), it seems natural that it would not influence molting. The manuscript can be better written to follow this logical line.

Reply: Thanks for reviewer’s valuable and thoughtful comments. According to previous report, CHS2 is mainly involved in the formation of chitin in peritrophic matrix (Tribolium chitin synthase genes TcCHS1 and TcCHS2 are specialized for synthesis of epidermal cuticle and midgut peritrophic matrix; The gene, expression pattern and subcellular localization of chitin synthase B from the insect Ostrinia furnacalis). Silencing of SlCHS2 by RNA interference has no significant influence for molting of S. litura. However, However, the specific reasons need further research.

  1. The final paragraph of the introduction needs to be re-written for improved clarity (L81-87), in particular, the following sentence (L85-86): “However, this silencing had no significant effect after knockdown of SlCHS2.”

Reply: Thanks for reviewer’s thoughtful suggestions. We have re-written the final paragraph of the introduction, seeing Line 92 to Line 101.

  1. This is confusing because in the results it says the following (L262-266): “After injection of dsSlCHS1, the S. litura larvae could not transform into pupa normally. In addition, the adults could not moult completely. However, the larvae could transform into pupae and adults after silencing SlCHS2 compared with the control groups (dsGFP treatment) (Figure 6C). These results suggested that knockdown of SlCHS1 inhibits the synthesis of chitin and influence S. litura moulting, while SlCHS2 has no significant effect on S. litura moulting.”

Reply: We are sorry for unclear descriptions in previous manuscript. We have revised the ambiguous descriptions, seeing Line 293 to Line 299.

  1. As written L85-86 would imply that knocking down SlCHS2 somehow negates the effects of silencing SlCHS1, which does not seem to be what the results are stating. I suspect this is a wording issue.

Reply: Thanks for reviewer’s valuable comments. According to previous reviewer’s suggestions, we have revised the related descriptions, seeing Line 97 to Line 101. In this study, we performed many times of repeated experiments for RNA interference. The results always showed that silencing of SlCHS1 influences S. litura larvae pupate and moulting. However, silencing of SlCHS2 has no significant influence for molting of S. litura. According to previous manuscript, many insect species encode two chitin synthase genes (CHS1 and CHS2). In Drosophia, the CHS1 gene is strongly expressed in stage 14-16 embryos at the time of peak embryonic cuticle deposition, while the highest levels of CHS2 gene expression can be found in both larval and adult hindguts and trachea (Genetic control of cuticle formation during embryonic development of Drosophila melanogaster). It indicated that CHS1 and CHS2 gene might play different role in the formation of chitin. However, we can’t be sure whether knocking down SlCHS2 somehow negates the effects of silencing SlCHS1.    

  1. Finally, the developmental stage analyses don’t seem to tie back to the objectives of the work. Are certain stages more important for pest control? How does this relate their fourth instar findings? SlCHS1 and SlCHS2 follow the same pattern, so I am not sure what the take away is.

Reply: Thanks for reviewer’s thoughtful comments. We considered that the fourth-instar is a critical stage for growth and development of S. litura. Before the fourth-instar stage, the growth of S. litura larvae is relatively slow to satisfy the development of various tissues, and then the larvae enter the glutonous stage. During this stage, the integument and peritrophic matrix enter a rapid growth stage. Therefore, the S. litura can promote chitin synthesis by increasing the expression of SlCHS1 and SlCHS2 to maintain the structure of integument and peritrophic matrix. However, the specific reasons need further research.

Reviewer 3 Report

In their manuscript, Yu et al. describe and characterize two chitin synthase (CHS) genes in the common cutworm Spodoptera litura.  As the last step in the chitin biosynthesis pathway, CHS is a potential target for control strategies and its characterization in various insect species represents a worthy goal. In addition to bioinformatic analyses, functional knock-down experiments are critical for implicating specific genes in particular aspects of organismal biology.

I found the manuscript well written and the introduction comprehensive. The results generally indicate tissue- and developmental-stage specific expression of CHS1 and CHS2 in the cutworm and implicate 20E in gene regulation. Knockdown experiments suggest that CHS1 but not CHS2 may be involved in moulting. There are several aspects of the materials and methods that must be improved before publication. Furthermore, proper controls are lacking for the FISH data and this must be corrected or the FISH data removed altogether.

In summary, I find the results of Yu et al. to be relevant and interesting, but think that these major issues should be addressed before publication. Please see my specific comments below.

Major comments:

– line 103: why were DMSO control injections performed at a different stage than the treatment injections?  This discrepancy must be explained.

– line 123: the description of the BLAST approach is not comprehensive and must be expanded upon.  What parameters were used? Why was the D. citri database queried instead of S. litura? What version of the genome assembly or transcriptome was used as the database? Were there any other orthologues identified in the BLAST approach, and how were these two candidates decided upon?

– the phylogenetic tree reveals several species that have only a single CHS gene (either CHS1 or CHS2).  This could be due to true gene loss, or could be due to incomplete genome assemblies or geneset annotations.  The authors should describe any efforts that were undertaken to investigate which of these outcomes are more likely to be true.

– It is difficult to evaluate the spread of the data in the RT-qPCR graphs in Figure 4 and Figure 5.  Ideally, the data should not be presented as bar plots, but as boxplots with all individual data points overlaid. 

– It appears from the data in Figure 6 that injection of 300ng/µL dsRNA against either CHS1 or CHS2 actually resulted in an increase in mRNA expression.  This is surprising, and should be discussed.  Is there precendence for this dose-dependent inversion of effect sign in the literature?  Were there observable phenotypes in adults under both conditions, or only at the higher doses?

– the FISH images are extremely difficult to evaluate.  The individual channels should be presented in black and white to enhance contrast, and there should be negative controls (e.g. sense probes, scrambled probes) to determine whether or not the expression is truly above background. The scale bars are also difficult to resolve.  If these deficiencies cannot be resolved, I would remove this data from the paper. The molecular analyses are sufficient to indicate tissue-specific expression and it is not clear what the FISH data would add to the story.

Minor comments:

– line 92 & 95: the composition of the ‘artificial diet’ should be defined.

– line 165: the percentage of triton x-100 in the PBST should be given

– line 216: it is not clear from the text or the plot which developmental stage was used for the tissue-specific RT-qPCR experiments. The methods suggest that it was 6th-instar larvae, but this should be clearly stated in the plot and in the legend.

– line 239: were there any phenotypes associated with the 20E-upregulation of SlCHS1 and SlCHS2?

– several typos (e.g. line 38 ‘control pest’; line 123 ‘quires’; line 162 ‘Rochen’; line 316 ‘northern bolt’; 

Author Response

  1. line 103: why were DMSO control injections performed at a different stage than the treatment injections? This discrepancy must be explained.

Reply: We are sorry for the incorrect descriptions in previous manuscript. The first day of sixth-instar larvae were selected for 20E and DMSO treatment. We have revised the related information, seeing Line 117.

  1. line 123: the description of the BLAST approach is not comprehensive and must be expanded upon. What parameters were used? Why was the citri database queried instead of S. litura? What version of the genome assembly or transcriptome was used as the database? Were there any other orthologues identified in the BLAST approach, and how were these two candidates decided upon?

Reply: Thanks for reviewer’s valuable comments. We have added the detailed descriptions in previous manuscript, seeing Line 136 to Line 145.

  1. the phylogenetic tree reveals several species that have only a single CHS gene (either CHS1 or CHS2). This could be due to true gene loss, or could be due to incomplete genome assemblies or geneset annotations. The authors should describe any efforts that were undertaken to investigate which of these outcomes are more likely to be true.

Reply: Thanks for reviewer’s thoughtful comments. CHS1 and CHS2 amino acid sequences of other insect species were retrieved from NCBI database and related references. However, some insect species only contain a single CHS gene. We speculated that gene loss could be due to incomplete genome assemblies. Before construct the evolutionary tree, all CHS amino acid sequences were performed multiple sequence alignment. In addition, we have added an additional phylogeny tree to test the same relationships using maximum likelihood.

  1. It is difficult to evaluate the spread of the data in the RT-qPCR graphs in Figure 4 and Figure 5. Ideally, the data should not be presented as bar plots, but as boxplots with all individual data points overlaid.

Reply: Thanks for reviewer’s thoughtful suggestions. We have revised the Figure 3 in previous manuscript, seeing Line 261.

  1. It appears from the data in Figure 6 that injection of 300ng/µL dsRNA against either CHS1 or CHS2 actually resulted in an increase in mRNA expression. This is surprising, and should be discussed. Is there precendence for this dose-dependent inversion of effect sign in the literature? Were there observable phenotypes in adults under both conditions, or only at the higher doses?

Reply: Thanks for reviewer’s valuable and valuable comments. In this study, prepupa S. litura larvae were selected for RNA interference to increase the efficiency. For RNA interference, a total of 3 μg dsRNA was injected into the larvae. In previous report, Ji et al. performed RNA interference to silence NADPH-Cytochrome P450 Reductase gene using fifth-instar S. litura larvae, and a total of 3 μg dsRNA was injected (Tobacco Cutworm (Spodoptera Litura) Larvae Silenced in the NADPH-Cytochrome P450 Reductase Gene Show Increased Susceptibility to Phoxim). In addition, we also found that injection of high concentration of dsRNA will cause a higher mortality. RT-qPCR results showed that injection of 300ng/µL dsRNA against either CHS1 or CHS2 actually resulted in an increase in mRNA expression. We considered that it might be due to the stress response. However, the specific reasons need further research.

  1. the FISH images are extremely difficult to evaluate. The individual channels should be presented in black and white to enhance contrast, and there should be negative controls (e.g. sense probes, scrambled probes) to determine whether or not the expression is truly above background. The scale bars are also difficult to resolve. If these deficiencies cannot be resolved, I would remove this data from the paper. The molecular analyses are sufficient to indicate tissue-specific expression and it is not clear what the FISH data would add to the story.

Reply: Thanks for reviewer’s valuable suggestions. We have deleted the FISH results, seeing Line 313

  1. line 92 & 95: the composition of the ‘artificial diet’ should be defined.

Reply: Thanks for reviewer’s valuable comments. We have added the related descriptions about the main component of the artificial diet in previous manuscript, seeing Line 107 to Line 108.

  1. line 165: the percentage of triton x-100 in the PBST should be given

Reply: Thanks for reviewer’s valuable suggestions. We have deleted the related descriptions for FISH experiment, seeing Line 180 to Line 195.

  1. line 216: it is not clear from the text or the plot which developmental stage was used for the tissue-specific RT-qPCR experiments. The methods suggest that it was 6th-instar larvae, but this should be clearly stated in the plot and in the legend.

Reply: Thanks for reviewer’s valuable and thoughtful comments. We have added the detailed information in previous manuscript, seeing Line 246 to Lin249 and Line 263.

  1. line 239: were there any phenotypes associated with the 20E-upregulation of SlCHS1 and SlCHS2?

Reply: Thanks for reviewer’s valuable comments. After injection of 20E, we did not observe phenotypic changes associated with the 20E-upregulation of SlCHS1 and SlCHS2

  1. several typos (e.g. line 38 ‘control pest’; line 123 ‘quires’; line 162 ‘Rochen’; line 316 ‘northern bolt’;

Reply: Thanks for reviewer’s valuable suggestions. We have revised several typos in previous manuscript.

Round 2

Reviewer 3 Report

The authors have significantly improved the manuscript based on reviewer's concerns.

Major comment: The description of the statistical tests used must be detailed for Figures 3, 4, and 5.  What test was performed in SPSS? t-test? wilcoxon? etc.?

Author Response

Major comment: The description of the statistical tests used must be detailed for Figure 3, 4, and 5. What test was performed in SPSS? t-test? wilcoxon? etc?

Reply: Thanks for reviewer’s valuable and thoughtful comments. All obtained data in this study are presented as the means ± standard of three replicates and were analyzed using one-way analysis of variance (ANOVA) and Tukey’s test. We have added the detailed information for Figure 3, 4 and 5 in previous manuscript, seeing Line 273, Line 291 and Line 311.